# Effect of Defects on Spontaneous Polarization in Pure and Doped LiNbO_3_: First-Principles Calculations

**DOI:** 10.3390/ma12010100

**Published:** 2018-12-29

**Authors:** Weiwei Wang, Dahuai Zheng, Mengyuan Hu, Shahzad Saeed, Hongde Liu, Yongfa Kong, Lixin Zhang, Jingjun Xu

**Affiliations:** 1School of Physics, Nankai University, Tianjin 300071, China; weiweiwang@mail.nankai.edu.cn (W.W.); 2120170171@mail.nankai.edu.cn (M.H.); shehzadsaeed2003@yahoo.com (S.S.); liuhd97@nankai.edu.cn (H.L.); lxzhang@nankai.edu.cn (L.Z.); jjxu@nankai.edu.cn (J.X.); 2The MOE Key Laboratory of Weak-Light Nonlinear Photonics and TEDA institute of Applied Physics, Nankai University, Tianjin 300457, China

**Keywords:** lithium niobate, spontaneous polarization, dipole moment, defects

## Abstract

Numerous studies have indicated that intrinsic defects in lithium niobate (LN) dominate its physical properties. In an Nb-rich environment, the structure that consists of a niobium anti-site with four lithium vacancies is considered the most stable structure. Based on the density functional theory (DFT), the specific configuration of the four lithium vacancies of LN were explored. The results indicated the most stable structure consisted of two lithium vacancies as the first neighbors and the other two as the second nearest neighbors of Nb anti-site in pure LN, and a similar stable structure was found in the doped LN. We found that the defects dipole moment has no direct contribution to the crystal polarization. Spontaneous polarization is more likely due to the lattice distortion of the crystal. This was verified in the defects structure of Mg^2+^, Sc^3+^, and Zr^4+^ doped LN. The conclusion provides a new understanding about the relationship between defect clusters and crystal polarization.

## 1. Introduction

As an important functional material, lithium niobate (LiNbO_3_, LN) possesses a range of excellent properties, such as acousto-optic, ferroelectric, and photorefractive properties and it has been widely used in a series of nonlinear optical devices [1,2]. In these applications, defects play an important role, so the defect structure is a significant part of LN research. In order to explore the relationship between defects and crystal properties, there were many defect models presented in history. From literature and experimental data, the Li-vacancy model consists of niobium anti-site and Li vacancies are energetically more favorable [3,4,5]. For a Li-vacancy model, Kim et al. [6] proposed that a niobium anti-site should be surrounded by the three nearest-neighbor (NN) lithium vacancies and one additional lithium vacancy shifted along the *z* direction. The arrangements with the crystal bulk provide a defect dipole moment responsible for changing the ferroelectric polarization. Afterwards, Xu et al. [7] pointed out the configuration that four lithium vacancies all located at the NN sites possess the lowest energy. The non-uniaxial dipole moments associated with the defect clusters could affect the properties of the system locally. However, theoretical calculations and analysis revealed that the location of Li vacancies around the Nb anti-site has many other possibilities. Yet, there is no generally accepted geometry structure of the defect complex models.

In addition, the local environments around the dopants in LN have always been concerning. Various experimental techniques and theoretical calculations were applied to investigate the defect structure and explore the defect influence on crystal properties, such as the photorefraction, spontaneous polarization, and the optical and nonlinear optical properties [8,9,10,11,12]. Although some experimental results reflect the site selection of dopant ions, it is still difficult to display the relationship between defects and spontaneous polarization. As a ferroelectric material, spontaneous polarization is a key characteristic of LN that shows the potential of ferroelectric ceramics [13]. Exploring the effect of a defect structure on spontaneous polarization can help us better control the spontaneous polarization of crystals, and, thus, suppress additional light scattering, i.e., Barkhausen noise, and other adverse effects, to achieve quasi-phase matching and further expand the application band range of nonlinear crystals [14,15].

As we know, the defects deforming the surrounding lattice by capturing or losing electrons would affect the normal site of Li and Nb ions. A dipole moment will be formed because of the positive and negative defect pair, which would influence the crystal polarization in the LN crystal. In this work, we concentrated on the defect dipole moment and explored its role in crystal polarization. Based on the first-principle, we adopted the density functional theory (DFT) to explore the configurations of the defect complex and the dipole moment formed by defects in pure and doped LN. Then we found the relationship of defects and spontaneous polarization through the direction and contribution analysis of the dipole moment in the crystal.

## 2. Methods

Our calculation was performed using the Vienna *ab initio* Simulation Package (VASP) code [16,17] with the projector-augment-wave (PAW) pseudopotentials [18,19]. The structures were modeled on the basis of the general gradient approximation (GGA) [20]. Due to the standard PAW potentials, the Li 2*s*^2^ Nb *4p*^6^*4d*^2^*5s*^1^, O *2s*^2^*2p*^4^ states were treated as valence electrons. Electronic wave functions were expanded in a plane wave basis set with an energy cutoff of 400 eV [21]. A hexagonal supercell containing 540 atoms for calculating intrinsic defects and 240 atoms for calculating extrinsic defects with a 2 × 2 × 1 k-points mesh over the Brillouin zone scheme are used for the perfect crystal and defect calculations [22]. The structure is obtained from the full optimization of the original cell reference to the experimental lattice parameters, and the error in the repeat optimization process is less than one-thousandth. In addition, the structural relaxation was performed using 0.01 eV/Å as the force convergence criterion.

It is known that the total internal energies obtained from DFT calculations correspond to the Helmholtz free energy at a zero temperature. Considering the temperature difference between the VASP work environment and the real condition, we had to think about the zero-point vibrations in free energy. However, experimental and theoretical aspects indicated that, at the room temperature of 300 K, the entropies of point defects fall about 0 to 0.26 eV. It seems too small to consider compared with the formation energy of crystals [23,24,25]. Nonetheless, strain effects can be considered negligible in large cell. The electrostatic interactions between periodic images mirror real interactions between the defect clusters in samples, which are caused by the periodic boundary conditions [23,26,27]. We obtained the total energy and defect formation energies (DFEs) from a DFT calculation. Generally, DFEs of a defect or defects *X* with charge q can be calculated using the equation below [28,29].
(1)$$\Delta E_{f}(X^{q})=E^{total}(X^{q})-E^{total}(\mathrm{perfect})+\sum_{i}n_{i}u_{i}+q(E_{F}+E_{v}+\Delta V)$$
where *E^total^*(*X^q^*)** is the total energy of the supercell with defects. *E^total^*(perfect) is the total energy of the perfect supercell with no defects. *n_i_* is the number of atoms of species *i* that have been added or removed when the defects are created, and *μ_i_* indicates the corresponding chemical potential. *E_ν_* is the bulk valence band maximum (VBM), and *E_F_* is the Fermi level with regard to the VBM. Δ*V* aligns the reference potential in the defect supercell with that in the bulk [28].

According to the thermodynamic considerations to maintain the stability of LN, we have plotted the thermodynamically stable region of the LiNbO_3_ constituents to limit the possible range of *μ_i_* in Figure 1. The chemical potential of its components changed within a certain range in accordance with a different reference phase. As shown in Figure 1, the region enclosed between points A, C, E, and G corresponds to one of the extremes of LN, which is in equilibrium with niobium oxide while the triangle BDF satisfies the other extreme limit condition from the equilibrium equation of lithium oxide. The red region is an area that meets all restrictions, and line CE represents the Nb-rich condition. The line BF indicates the Li-rich condition. In our calculation, we employ the Nb_2_O_5_ reference to simulate the environment in LN. Thus, the chemical potentials of Li and Nb used here are −3.715 and −20.605 eV.

## 3. Results and Discussion

### 3.1. Intrinsic Defects in LN

So far, the most convincing intrinsic defect model for LN is the Li vacancy model, where the lithium vacancies (V_Li_^−^) are located around the niobium anti-site (Nb_Li_^4+^). In order to analyze the specific configuration of lithium vacancies, we initially need to consider their distribution. In Figure 2, the vicinity analysis of the lithium sub-lattice is shown. The red ball indicates the Nb_Li_^4+^, and its first neighbors (1NNs) of six atoms were represented by blue balls, which were marked with N11, N12, N13, N14, N15, and N16. The second nearest neighbors (2NNs), the third nearest neighbors (3NNs), and the fourth nearest neighbors (4NNs) are shown by the purple, green, and yellow balls, respectively. They are distinguished with the same method as the 1NNs. The top view of the vicinity analysis is shown in Figure 2b to display the positional relationship of these neighbors.

According to the neighborhood analysis, there are many possibilities for the four lithium vacancies around the Nb_Li_^4+^, and considering all possible arrangements is an arduous task. It can be seen from Figure 2 that the distance between the Nb_Li_^4+^ and its 4NNs is already 6.38 Å. The 5NNs are far from a reasonable distance. Taking into account the role of Coulomb force and the limit of the supercell, four lithium vacancies acting as 4NNs or 5NNs is beyond our consideration. Therefore, in our 540-atom supercell, lithium vacancies can be randomly distributed at the Nb inversion of 1NN, 2NN, 3NN, and 4NN. To simplify calculations and demonstrations, clusters are divided into the following six types.
four lithium vacancies as 1NNs;four lithium vacancies as 2NNs;four lithium vacancies as 3NNs;three lithium vacancies as 1NNs and one other NN;three lithium vacancies as 2NNs and one other NN;two lithium vacancies as 1NNs and two other NNs;

We take the (I) situation as an example to describe the specific calculation and analysis process. It can be seen from Figure 2 that there are six 1NNs around the Nb anti-site and they are not equivalent. This non-equivalent position results in at least 15 different configurations of the four lithium vacancies, and each type exists different degrees of degeneracy. From the literature before, it is difficult to find a clear relationship between the polarization and DFE [7]. Therefore, the cohesive energy and formation energy of defect clusters can be regarded as the main judgment for the most stable structure.

Similarly, we considered all II–VI situations and possible degeneracy conditions of the Nb_Li_^4+^ + 4V_Li_^−^ model to judge the most stable model in different situations. In each case, the difference in the formation energy of the different models is within 0.5 eV, and this small difference determines the stability of the model. The formation of the most stable Nb_Li_^4+^ + 4 V_Li_^−^ model in different situations ranged from I to VI and are listed in Table 1. The models of Kim [6] and Xu [7] based on our unit cell are listed for comparison. The polarization change of the special models are shown in the table too. In Table 1, we found that the model VI possesses the lowest formation energy. It is not an accidental result. In the IV situation, three lithium vacancies as 1NNs and one other NN. When the other NN is 2NN, the model has lower formation energy. For model VI, three lithium vacancies as 2NNs and one other NN in the model 2NN + 2NN + 2NN + 1NN is more stable. The configuration of these two models is similar to the most stable model 1NN + 1NN + 2NN + 2NN. This shows the rationality of the most stable structure.

Changes in the crystal structure introduced by the defects will affect the properties of the crystal, and the charged point defects will form an electric dipole moment inside of the crystal too. The positive charge center of the dipole moment is the Nb antisite. The negative charge center is determined by the position of the lithium vacancy and the direction of the dipole moment may cause a change in the polarization. We explored the specific structural information of the most stable model and the second most stable model, as shown in Figure 3. The red balls represent the site of positive charge center of the dipole moment. The negative charge centers are more complicated. In the most stable structure (Figure 3a), if a line is drawn from the Nb anti-site along the opposite direction of the *b* axis, then the two 2NNs are symmetric about this line. The intersection of the line with the connections of the two 1NN is the midpoint of the two 1NNs. The distribution of lithium vacancies has a high symmetry and the midpoint is the negative charge centers. The direction of the electric dipole moment usually points from the negative charge towards the positive charge. As such, the direction of the dipole moment along the b axis in the most stable model (see the green arrow in Figure 3a). The polarization change caused by the defect clusters is about 2.8 μC·cm^−2^. It is in well agreement with the data in Xu’s paper compared to the polarization of the LN system [7]. In the second most stable structure (Figure 3b), one of the 1NN is located symmetrically with b at the Nb anti-site with another 1NN. The two 2NNs are lying in the same sites with the most stable structure. In addition, the direction of the dipole moment due to intrinsic defects is also along the direction of the *b* axis, which is consistent with the most stable structure. The direction of these two defect dipole moments are both perpendicular in the direction of spontaneous polarization of the crystal (*c* axis). Since the non-polarized polarization caused by the dipole moment can be expected to cancel out in an ideal environment [7], the entire cluster of defects does not directly contribute toward the spontaneous polarization.

The spontaneous polarization along the c-axis direction appears in the LN crystal due to the Nb ions leaving the center of the oxygen octahedron and the Li atoms leaving the oxygen planes. The spontaneous polarization of CLN is different from that of the pure LN, which indicates that spontaneous polarization is influenced by the defects situation. However, the above results indicate that the direction of the dipole moment of intrinsic defect clusters is along the b direction and can be canceled out ideally. It has no direct effect on bulk polarization. It is in agreement with the experiment that there is no polarization component in the *a* or *b* axis. In addition, the polarization component in the *c* axis may be related to lattice distortion. The results provided a new understanding of the relationship between the polarization direction and defect models.

### 3.2. Divalent, Trivalent, and Tetravalent Ions Doped LN

From the above discussion, we can get that the intrinsic defects dipole moment have no direct contribution to the spontaneous polarization in pure LN. The spontaneous polarizations of the doped LN are different from the pure LN, Mg^2+^, Sc^3+^, and Zr^4+^ doped LN (LN:Mg, LN:Sc, and LN:Zr) as representations of divalent, trivalent, and tetravalent dopant ions are explored. Based on the previous experimental exploration and theoretical calculation, in Nb_2_O_5_-rich conditions, most dopants sit on the Li site in their highest charge state and the charge is compensated by lithium vacancies in a lower concentration [30,31,32,33]. Hence, one, two, and three lithium vacancies compensate Mg_Li_^+^, Sc_Li_^2+^, and Zr_Li_^3+^, respectively. The specific configurations of the defect pairs formed by divalent, trivalent, and tetravalent ions will be discussed in this part.

The situations of Mg_Li_^+^ + V_Li_^−^ are simple. The only lithium vacancy can be the 1NN, 2NN, or 3NN. In addition, the classifications for Sc_Li_^2+^ + 2V_Li_^−^ are not complicated. The combination of the two lithium vacancies are 1NN + 1NN, 1NN + 2NN, and 2NN + 2NN, as shown in Table 2. For tetravalent ions Zr_Li_^3+^ + 3V_Li_^−^, due to the limit of the supercell, the possible defect clusters will be divided into the following categories.
three lithium vacancies as 1NNs;two lithium vacancies as 1NNs and one other;two lithium vacancies as 2NN and one other;three lithium vacancies as 2NNs;

The most stable structure and its formation energy of each condition are listed in Table 2. In the most stable defect structures of Mg^2+^ and Sc^3+^, doped LN and lithium vacancies also prefer 2NNs. It is similar to the conclusions obtained in the pure LN. For Zr_Li_^3+^ + 3V_Li_^−^, not all lithium vacancies are located at 2NN and there are two. The slight difference in defect configuration of high-valence and lower valence ions doped LN indicates the difference in defect polarization and we will discuss it further below.

The most stable structure and dipole moment analysis of LN:Mg, LN:Sc, and LN:Zr are shown in Figure 4. It can be seen from Figure 4a, in Mg_Li_^+^ + V_Li_^−^, lithium vacancy is the 2NN site. It lies in the direction of the *a*-axis. Therefore, the direction of the defect dipole moment is along the opposite direction of the *a*-axis and perpendicular to the *c*-axis. It does not contribute to the spontaneous polarization of the crystal. As shown in Figure 4b, in the defect structure of LN:Sc, the direction of the defect dipole moment corresponds to the combination of the a-axis and the *b*-axis. It is orthogonal to the *c*-axis. The two models are in line with the intrinsic defect clusters we discussed above. The direction of the dipole moment are in a plane that is perpendicular to the *c*-axis, which can be expected to cancel out ideally. However, the defect cluster of Zr_Li_^3+^ + 3V_Li_^−^ is an exception. Seen from Figure 4c,d, the structure of LN:Zr consists of two second nearest neighbors in the same layer as the Zr_Li_^3+^ and one nearest neighbor in the lower layer. The dipole moment of the defect cluster is not parallel to the *a*, *b* plane but has a small angle to the plane, which is about 21 degrees. The dipole moment contributes to both the horizontal plane and the *c*-axis direction. The contribution in the horizontal plane along the black dotted line in Figure 4d, which can be canceled out in an ideal environment. Less contribution in the vertical direction along the *c*-axis coincides with the spontaneous polarization direction of the crystal. The polarization change caused by the defect clusters along the *c*-axis is about 1.41 μC·cm^−2^, which is pretty small compared to the polarization of congruent LN at about 71 μC·cm^−2^ [34]. The particularity of LN:Zr is that Zr is a high charge state ion, which may occupy the Nb position in the LN. The influence on Nb is intensified, which makes the relationship between defects and polarization complicated. From the current results, the dipole moment of the defects is not directly related to the polarization.

To summarize the stable defect structure models of Mg_Li_^+^ + V_Li_^−^, Sc_Li_^2+^ + 2V_Li_^−^, Zr_Li_^3+^ + 3V_Li_^−^, we found that lithium vacancies prefer to be 2NN rather than the nearest neighbors, which is similar to the structure of the intrinsic defect above. Neighborhood analysis enriches the structural type of the defect. In addition, polarization analysis shows that the direction of the dipole moment formed by the dopants is generally perpendicular to the direction of spontaneous polarization. High charge state dopants Zr show little influence on polarization. Considering that the valence and size of high charge state ions are closer to the Nb ions in the crystal, the influence of Zr on Nb ions will increase. This paper mainly considers the influence of Zr and lithium vacancies on polarization. The more complex mechanisms are for further study.

## 4. Conclusions

Based on the DFT calculation, the specific configuration of intrinsic defect clusters in pure and doped LN have been investigated. For pure LN, the energetically preferable cluster consisting of two 1NN lithium vacancies and two 2NN lithium vacancies of the Nb anti-site was selected among all possible defect cluster combinations. In the most stable model, the polarization direction of the defect cluster dipole moment was perpendicular to the bulk polarization direction (*c* axis), which indicated that the defects dipole moment had no direct contribution to the crystal polarization. We found that the lithium vacancy of LN:Mg, LN:Sc, and LN:Zr lies in 2NN rather than the 1NN, and the defect dipole moment almost had no direct contribution to the crystal polarization. We proposed that the crystal polarization difference does not originate from the dipole moment of dopants, but may be the lattice distortion of the crystals. All the results would give us a new understanding of intrinsic defect clusters, and would provide a new approach for the exploration of the relationship between dopants and properties.

## Figures and Tables

**Figure 1 materials-12-00100-f001:**
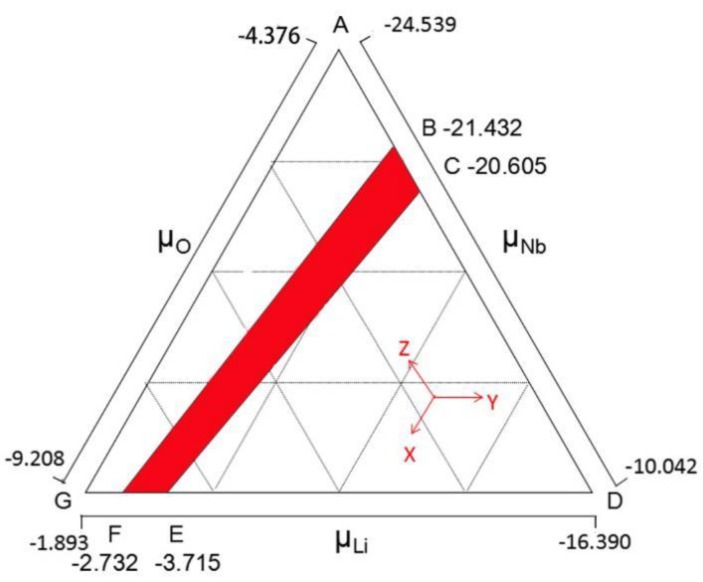
Stability range of chemical potentials of the elements in LiNbO_3_. The chemical potential in the red region satisfy all the limit conditions, and the line BF, CE represent the reference state Li_2_O and Nb_2_O_5_, respectively.

**Figure 2 materials-12-00100-f002:**
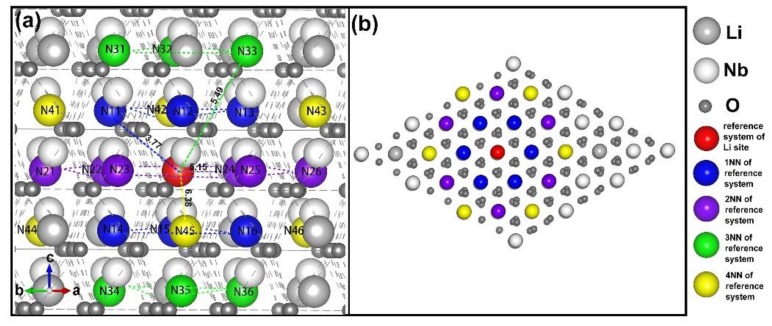
The vicinity analysis of the lithium (Li) sub-lattice. (**a**) The red ball present dopant in a Li sub-lattice. In addition, blue, purple, green, and yellow balls indicate the INN, 2NN, 3NN, and 4NN sites from the dopant site. The distance between the dopant site and the nearest neighbor sites are given. Its unit is Å. (**b**) is the top view of (**a**) along the opposite direction of the *c* axis.

**Figure 3 materials-12-00100-f003:**
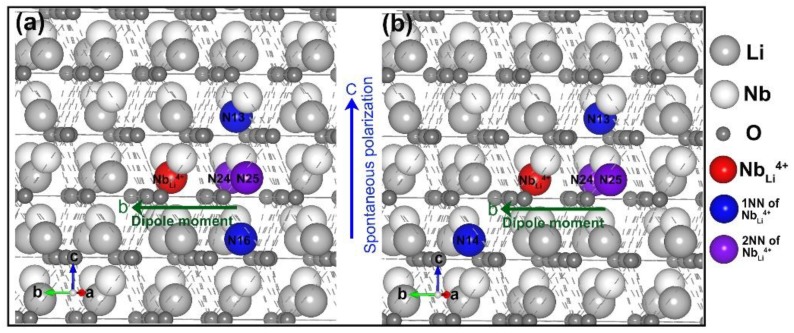
(**a**) The most stable model and (**b**) the second stable model of LN. The red balls present the Nb antisite. Blue and purple balls represent the lithium vacancies in 1NN and 2NN sites of Nb antisite, respectively. In addition, the blue arrow shows the direction of spontaneous polarization of the crystal, while the green arrow indicates the direction of the dipole moment.

**Figure 4 materials-12-00100-f004:**
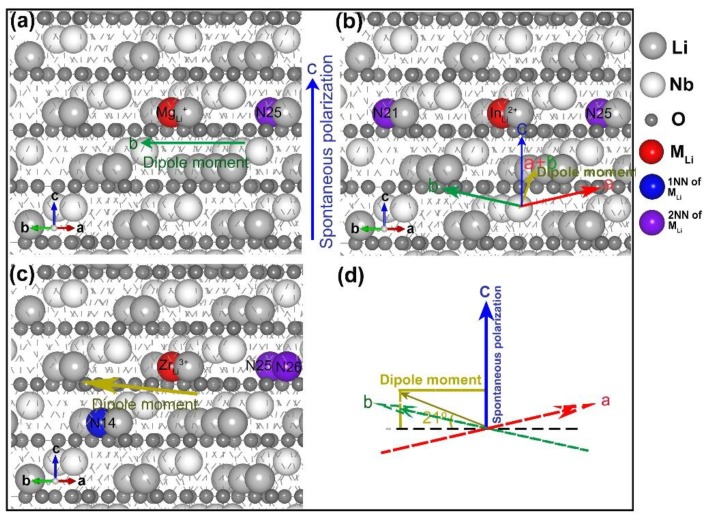
The most stable model of LN:Mg (**a**), LN:Sc, (**b**) and LN:Zr (**c**). The dopants ions Mg^2+^, Sc^3+^, and Zr^4+^ are represented by red balls in the figure, respectively. Blue and purple balls are lithium vacancies in 1NN and 2NN site of dopant ions. In addition, the blue arrow shows the direction of spontaneous polarization of crystal, while green arrows and yellow arrows indicate the direction of the dipole moment in (**a**–**c**). (**d**) is the amplification structure of the dipole moment. The red, green, and blue arrows represent the direction of the *a*, *b* and *c* axis in the crystal, and the yellow arrow shows the direction of the dipole moment.

**Table 1 materials-12-00100-t001:** The most stable Nb_Li_^4+^ + 4V_Li_^−^ model in different situations (I–VI) with the formation energy and specific configuration.

Situations	DFE (eV)	Configuration of 4 Lithium Vacancies	Polarization Change (μC·cm^−2^)
I four lithium vacancies as 1NNs;	13.280	1NN + 1NN + 1NN + 1NN	
II four lithium vacancies as 2NNs;	13.204	2NN + 2NN + 2NN + 2NN	
III four lithium vacancies as 3NNs;	13.357	3NN + 3NN + 3NN + 3NN	
IV three lithium vacancies as 1NNs and one other NN;	13.246	1NN + 1NN + 1NN + 2NN	
V three lithium vacancies as 2NNs and one other NN;	13.222	2NN + 2NN + 2NN + 1NN	
VI two lithium vacancies as 1NNs and two other NNs;	13.166	1NN + 1NN + 2NN + 2NN	2.8
Kim’s Model	13.389	1NN + 1NN + 1NN + one in the direction of polarization	3.2 [7]
Xu’s Model	13.307	1NN + 1NN + 1NN + 1NN	0.8 [7]

**Table 2 materials-12-00100-t002:** Formation energies of specific configurations of Mg_Li_^+^ + V_Li_^−^, Sc_Li_^2+^ + 2V_Li_^−^, and Zr_Li_^3+^ + 3V_Li_^−^.

Mg_Li_^+^ + V_Li_^−^		Sc_Li_^2+^ + 2V_Li_^−^		Zr_Li_^3+^ + 3V_Li_^−^	
Specific Configuration	DFE (eV)	Specific Configuration	DFE (eV)	Specific Configuration	DFE (eV)
1NN	0.924	1NN + 1NN	6.575	1NN + 1NN + 1NN	9.343
2NN	0.862	1NN + 2NN	6.508	1NN + 1NN + 2NN	9.338
3NN	0.967	2NN + 2NN	6.466	1NN + 2NN + 2NN	9.111
				2NN + 2NN + 2NN	9.133

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
