# Peer review of "Effect of Defects on Spontaneous Polarization in Pure and Doped LiNbO3: First-Principles Calculations"

_materials, 2018, doi:10.3390/ma12010100_

Reviewer 1 Report

Dear Author,

This paper deals with the study based on DFT calculation of intrinsic defects of pure and doped lithium niobate crystals. It is also shown that the Li vacancy lies in 2NN and that the defect dipole moment has no directly contribution to the crystal polarization for Sc, Zr and Mg doped LN crystals. It is so suggested that the crystal polarization is due to the lattice distortion.

The paper is original and can bring new ideas for understanding intrinsic defect clusters of LN crystals so I recommend it's publication.

Author Response

Response to Reviewer 1 Comments

Point 1: This paper deals with the study based on DFT calculation of intrinsic defects of pure and doped lithium niobate crystals. It is also shown that the Li vacancy lies in 2NN and that the defect dipole moment has no directly contribution to the crystal polarization for Sc, Zr and Mg doped LN crystals. It is so suggested that the crystal polarization is due to the lattice distortion.

The paper is original and can bring new ideas for understanding intrinsic defect clusters of LN crystals so I recommend it's publication.

Response 1: Thank you very much for your reviewing processes and the positive comments.

Reviewer 2 Report

It is a nice Ms which may be published after some minor revisions:

1.       The authors are advised to formulated briefly the innovation of the work with respect to  many other in the field.

2.       More attention should be devoted to the stability of the samples in time.

3.       In the reference/bibliographic part the authors should emphasize that for the optical and nonlinear optical properties of the of LiNbO3 the intrinsic defects following the  band structure  calculations should  play a crucial role as well as phonon anharmonic contributions [Non-Stoichiometric Defects and Optical Properties in LiNbO3.//Journal of Physical Chemistry.B,V. 105, N 48, (2001), pp. 12242-12248 .

4.       The Conclusions should be more sound.

5.       Some efforts should be devoted to style improvement of the MS

Author Response

Response to Reviewer 2 Comments

Dear Reviewer,

Thank you for the reviewing processes and kind comments and suggestions. We revised the manuscript in accordance with your suggestions (“Track Changes” function). In the following, we list the responses and the summary of the changes.

Point 1: The authors are advised to formulated briefly the innovation of the work with respect to many other in the field.

Response 1: Thank you very much for your kind suggestion. Various experimental techniques and theoretical calculation were applied to investigate the defect structure of dopants. In this work, we concentrate on finding the relationship of defects dipole moment and spontaneous polarization in the pure and doped LN crystal. The results give us a new understanding of intrinsic defect clusters, and provide a new approach for the exploration of the relationship between dopants and properties. Please check the details in the introduction and conclusion sections of the revised manuscript.

Point 2: More attention should be devoted to the stability of the samples in time.

Response 2: It is appreciated for your kind suggestion. In order to get the most stable structure model, we tried to optimize the LN cells under different lattice parameters referencing the experimental results. We repeat the optimization process of the original cell, and the error is less than one thousandth. Details were refined in the Method section of the revised manuscript.

Point 3: In the reference/bibliographic part the authors should emphasize that for the optical and nonlinear optical properties of the of LiNbO3 the intrinsic defects following the band structure calculations should play a crucial role as well as phonon anharmonic contributions [Non-Stoichiometric Defects and Optical Properties in LiNbO3.//Journal of Physical Chemistry.B,V. 105, N 48, (2001), pp. 12242-12248.

Response 3: Thanks so much for your kind suggestion. We supplemented some related references about the optical and nonlinear optical properties of LiNbO3 exploring the relationship between the intrinsic defects and properties of crystals, the above literature included. Please check the details in the revised manuscript.

Point 4: The Conclusions should be more sound.

Response 4: It is appreciated your kind comment. We optimized our conclusions based on the calculation results that “two nearest neighbors and two second nearest neighbors of Nb antisite” was the energetically preferable cluster among all possible defect cluster combinations, and the defects dipole moment had no directly contribution to the crystal polarization. Please check the details in the revised manuscript.

Point 5: Some efforts should be devoted to style improvement of the MS.

Response 5: Thank you very much for your review work. The style of the MS was further improved from the beginning to the end. The details were red sentenced in the revised manuscript.

Reviewer 3 Report

Manuscript ID:  materials-413164

Title:

Effect of Defects on Spontaneous Polarization in Pure and Doped LiNbO3: First-Principles Calculations

Authors:

Weiwei Wang, Dahuai Zheng, Mengyuan Hu, Shahzad Saeed, Hongde Liu, Yongfa Kong, Lixin Zhang, Jingjun Xu

Authors investigated the structure of characteristic defect of congruent LiNbO3. Their calculation based on DFT. The authors gave a probable, energetically favored configurations of the lithium vacancies around niobium antisite and di-, tri-, tetravalent dopant ions and they gave dipolmoment analysis as well.

The results of these calculations can be accepted, but the main objection concerning this work is the lack of the discussion and comparison. How these results agree with data of similar calculations in the literature? It would be worth to show a comparison table. Is it possible to explain the experimental results on the basement of these calculations?

Author Response

Dear Reviewer,

Thank you for the reviewing processes and kind comments and suggestions. We revised our manuscript in accordance with your suggestions (“Track Changes” function). In the following, we list the responses and the summary of the changes.

Point 1: The results of these calculations can be accepted, but the main objection concerning this work is the lack of the discussion and comparison.                                                                                    

Response 1: It is appreciated for your kind comments. We supplemented moderately discussion about our results and some references for comparison in the revised manuscript. Please check the details in revised manuscript.

Point 2: How these results agree with data of similar calculations in the literature? It would be worth to show a comparison table.

Response 2: Thanks for your kind inquiry and suggestions. We added the ways of modelling and the polarization change caused by the defect clusters in the Table I for simple comparison, instead of a new comparison table. There was an appropriate analysis of the table, then details can be checked in the revised manuscript.

Point 3: Is it possible to explain the experimental results on the basement of these calculations?

Response 3: Thanks for your question. The intention of the calculation is to provide the theoretical support for the experiment results or predict a new phenomenon. It makes sense that there is no additional polarization along the a and b axes in LN crystals. Our results also introduce a new way to explore the relationship between structure and properties. On the other hand, the results guide researchers to investigate the effects on spontaneous polarization from lattice distortion caused by defects in doped LN. In a word, these calculations would be contributed to understand the experimental results.

Materials EISSN 1996-1944 Published by MDPI AG, Basel, Switzerland RSS E-Mail Table of Contents Alert
Back to Top